# Magnetically Guided Micromanipulation of Magnetic Microrobots for Accurate Creation of Artistic Patterns in Liquid Environment

**DOI:** 10.3390/mi11070697

**Published:** 2020-07-18

**Authors:** Xingfu Li, Toshio Fukuda

**Affiliations:** 1Chongqing Key Laboratory of Manufacturing Equipment Mechanism Design and Control, Chongqing Technology and Business University, Chongqing 400067, China; 2Beijing Advanced Innovation Center for Intelligent Robots and Systems, Beijing Institute of Technology, Beijing 100081, China; tofukuda@nifty.com

**Keywords:** micromanipulation, magnetic guidance, magnetic microrobot, dot-matrix magnetic flux density, tissue engineering

## Abstract

In this paper, a magnetically guided micromanipulation method is proposed to accurately create artistic patterns with magnetic microrobots in a liquid environment for tissue engineering. A magnetically guided device is developed depend on symmetrical combination of square permanent magnets and array layout of soft magnetic wires, which changed the space distribution of magnetic field of conventional permanent magnet and generated powerful magnetic flux density and high magnetic field gradient. Furthermore, the morphological structure of the magnetic microrobot is flexibly adjusted via precise control of the volumetric flow rates inside the microfluidic device and the magnetic nanoparticles are taken along to enable its controllability by rapid magnetic response. And then, the spatial posture of the magnetic microrobot is contactless controlled by the magnetically guided manipulator and it is released under the influence of surface tension and gravity. Subsequently, the artistic fashions of the magnetic microrobots are precisely distributed via the dot-matrix magnetic flux density of the magnetically guided device. Finally, the experimental results herein demonstrate the accuracy and diversity of the pattern structures in the water and the developed method will be providing a new way for personalized functional scaffold construction.

## 1. Introduction

Robotic manipulation is a promising technology for medical engineering and the accurate operation reduces the burden of medical staff and increases work efficiency [1,2,3,4]. The different morphological and functional microrobots that are micro manufactured can be used as the cell carrier for specific scaffold construction in tissue engineering and the functional scaffolds constructed with microrobots can be used for repairing damaged tissues and replacing damaged organs in regenerative medicine [5,6]. Furthermore, advanced microrobot technology provides a technical support for reducing the pain of the patient in medical rehabilitation and the manipulation technology of the microrobots has been gradually developed for biomedical engineering [7,8,9]. However, there are still some challenges for tissue engineering in the drive and control of microrobots with different cells by micromanipulation in situ environment.

The design of the microrobots can draw inspiration from nature and different ways of controlling the movement of the microrobots in a liquid environment can be used [10,11,12]. Conventional methods attempt to build the complex microrobot with its own propulsion and navigation but this kind of robot is difficult to micromanufacture [13,14]. A more superior method creates the microrobot that responds to the outside environment and guide controls the movement of the microrobot from the outside field. This method includes two directions with contact operation and indirect micromanipulation. The intrusive operation has high precision but large destructiveness [15]. Correspondingly, the non-contact micromanipulation has small destructiveness but poor controllability [16,17,18].

An excellent method that the magnetic material responds to a magnetic field has been developed for non-contact micromanipulation of the microrobot and the movement of the microrobot could be controlled by the magnetic guide with the outside magnetic field [19,20,21,22]. In tissue engineering, magnetically guided micromanipulation of a magnetic microrobot has been researched in stem cell transplantation [23], development of biodegradable scaffolds [24], fabrication of microvascular-like structures [25], fiber functionalization [26], intracellular manipulation and measurement [27] and carrying and delivering targeted cells [28]. In biomedical engineering, magnetically guided micromanipulation was used for adaptive locomotion [29], superior performances in both wet and dry conditions [30], multimodal locomotion [31] and programmable three-dimensional magnetization and motions [32]. In medical engineering, magnetically guided micromanipulation was used for imaging-guided therapy [33], inaccessible lesions and healthcare [34], gastrointestinal tract exploration [35], gastrointestinal ultrasound [36], targeted pathogen killing [37], potential targeted immunotherapeutic applications [38] and stomach capsule endoscopy application [39]. However, with these methods, some problems are encountered when attempting to accurately micromanipulate the magnetic microrobots in a liquid environment, such as weak magnetic forces, difficult-to-control magnetic field distribution and difficulty in release after capture. In particular, it is difficult to overcome fluid disturbance and accurately construct stable patterns with microrobots for tissue engineering.

In this study, we develop a method to accurately create artistic patterns with magnetic microrobots in a liquid environment for personalized functional scaffold construction in tissue engineering. This method is accomplished by magnetically guided micromanipulation of the microrobot, which depends on the precise control of the magnetically guided manipulator combined with the orderly distribution of the magnetically guided device. To change the space distribution of magnetic field of the conventional permanent magnet, we create a magnetically guided device with the dot-matrix magnetic flux density which exist powerful magnetic flux density and high magnetic field gradient by the array layout of the soft magnetic wires to gather and discretize the magnetic field density. To add to the controllability of the microrobot, we adjust the morphology of the microrobot via precise control of the volumetric flow rates and employ magnetic nanoparticles to enable its rapid magnetic response. To prevent destructive manipulation, we apply a magnetically guided manipulator to quickly capture the microrobot and release it with the surface tension and gravity. To overcome fluid disturbance, we methodically navigate the microrobots to desired places and organize them to form diverse artistic patterns by the dot-matrix magnetic flux density in water. This magnetically guided micromanipulation provides a new method for personalized functional scaffold with orderly construction in tissue engineering and it provides a new approach for precise operation with non-invasive, non-destructive and non-contact in biomedical engineering.

## 2. Equipment and Methods

### 2.1. Micromanipulation Robot System Setup

The micromanipulation robot system is mainly composed of four subsystems, which can be provided by an experimental platform for accurate creation of diverse artistic patterns with magnetic microrobots. The schematic diagram is shown in Figure 1a. The sensor subsystem mainly comprised a stereomicroscope (S9 D, Leica Microsystems Inc., Heerbrugg, Switzerland), an industrial camera (MC170 HD, Leica Microsystems Inc., Heerbrugg, Switzerland) and an observation feedback interface. The control subsystem mainly comprised a stepping motor motion controller (ESP301-3N, Newport, Inc. Evry CEDEX, France), a piezometer motion controller (Model 8742, Newport, Inc. Irvine, CA, USA), a computer and a motor control interface. The drive subsystem mainly comprised three syringe pumps (Legato 111, KD Scientific, Inc. Holliston, MA, USA), a micromanipulation robot (UTS50PP, Newport, Inc. Evry CEDEX, France) and a magnetically guided manipulator which is composed of a micromanipulation robot (9063-XYZ-PPP-M, Newport, Inc. Irvine, CA, USA), a micro permanent magnet and a vertical lifting table. The mechanical subsystem mainly comprised a microfluidic device and a magnetically guided device and the partial enlarged drawing is shown in Figure 1b. The magnetically guided device is composed of 81 soft magnetic iron wires (SMIWs), a polydimethylsiloxane (PDMS) fixing module, a petri dish, five square permanent magnets and a plastic fixing bracket and the schematic diagram is shown in Figure 1c.

### 2.2. Magnetically Guided Micromanipulation Method

First, the magnetic microrobot was guided to the specific location above the magnetically guided device by the horizontal movement of the magnetically guided manipulator. The schematic diagram of this step is shown in Figure 2a. Secondly, the magnetic microrobot was lifted up to the liquid level with the vertical movement of the magnetically guided manipulator. The schematic diagram of this step is shown in Figure 2b. Thirdly, the magnetic microrobot was broken away from the constraint of the magnetic force due to the liquid surface tension and gravity and then it dropped due to gravity. The schematic diagram of this step is shown in Figure 2c. Subsequently, the path planning of the magnetic microrobot was completed by the piecewise fitting method and then the middle of the magnetic microrobot was captured and guided to the target location by the dot-matrix magnetic flux density. By using the above method, the magnetic microrobot was spatially organized into desired places to form an artistic pattern that depended on the precise control of the magnetically guided manipulator combined with the orderly distribution of the magnetically guided device. The schematic diagram of this step is shown in Figure 2d.

## 3. Results and Discussion

### 3.1. Micromanufacture of the Magnetically Guided Device

The magnetic flux density of the geomagnetic field is only 0.047 mT in Chongqing, China, so its contribution can be ignored in the calculation. The square permanent magnet was employed as the magnetic field source for micromanipulation and the length, width, height and magnetization intensity are 15, 15, 15 mm and 750,000 A/m, respectively. The distribution of the magnetic flux density on the surface could be simulated with the software package COMSOL multiphysics 5.0 and the minimum and maximum values of the magnetic flux density were found to approach 411 and 907 mT on the surface, respectively. The simulation result of the surface of the square permanent magnet for the vertical view when the magnetic pole of the square permanent magnet is up (or down) is shown in Figure 3a and the simulation result which through the center of the square permanent magnet for sectional view is shown in Figure 3b. The simulation results show that the edges of the square permanent magnet exhibit a powerful magnetic flux density and inhomogeneous magnetic field gradient but the center of the square permanent magnet exhibits a weak magnetic flux density and small magnetic field gradient. Regrettably, this contradicts what we expect to micromanipulate the magnetic microrobot in a central region of three-dimensional space which under the microscope.

In order to strengthen the magnetic flux density of the center of the micromanipulation space, we added four square permanent magnets, which surround the central square permanent magnet. Four square permanent magnets were symmetrically combined and their bottoms were flush with the bottom of the central square permanent magnet. The length, width and height of the four surrounded square permanent magnets are 30, 15 and 15 mm, respectively. The magnetization direction along the length of the magnet and the magnetization intensity is 750,000 A/m. The appearance when the magnetic pole of the central square permanent magnet is up (or down) and the magnetic pole of the four surrounding square permanent magnets are outward (or inward) and the simulation result of the surface of five square permanent magnets for the vertical view are shown in Figure 3c. The minimum and maximum values of the magnetic flux density were found to approach 671 and 1589 mT on the surface of the central square permanent magnet, respectively. Furthermore, we chose the sectional view which through the center of the central square permanent magnet and two centers of two symmetrically combined square permanent magnets and it is shown in Figure 3d. The simulation results show that the surface of the central square permanent magnet had enhanced.

During the experiment, when five square permanent magnets were placed under the petri dish, magnetic microrobots were dispersed and attracted to the bottom of the petri dish because of the powerful magnetic flux density of the central square permanent magnet. But when the magnetic microrobot was falling down from the liquid level, it easily escaped from the center to the edges of the central square permanent magnet because of the large magnetic flux density and the corresponding strong magnetic force located on the edges. Therefore, in order to improve the accuracy of microrobot localization, it is necessary to overcome the influence of the fluid disturbance by changing the distribution in three dimensions of the magnetic field of the square permanent magnet.

We needed to design a magnetically guided device that displays the powerful magnetic flux density and the homogeneous magnetic field gradient. Especially, the soft magnetic wire has an excellent performance of aggregating magnetic field density and we chose soft magnetic wires to further enhance the magnetic flux density above the surface of the central square permanent magnet. Furthermore, we design the array layout of the soft magnetic wires to disperse the magnetic flux density of the surface of the central square permanent magnet.

Firstly, because the center of the central square permanent magnet has relatively uniform magnetic flux density, we chose a rectangular area with a side length of 10 mm and located it in the optimal area for micromanipulation. Because the size of the magnetic microrobot and the limiting distance of the cell supply are less than 300 μm, we chose 81 SMIWs with 200 μm diameters and a separation distance of 500 μm. Secondly, to enhance the magnetic flux density and magnetic field gradient, the simulation result of SMIWs with same height is shown in Figure 4a. Therefore, appropriate height range of the magnetically guided array is 4~7 mm, specially, we chose the height of the SMIWs to be 5.5 mm. A dot-matrix magnetic flux density was obtained by the grid distribution of the SMIWs inside the magnetically guided device and the relatively homogeneous magnetic flux density was discretely distributed on the surface of the SMIW. Then, we simulated the magnetic flux density of the surface of the SMIWs, which is shown in Figure 4b. The minimum and maximum values of the magnetic flux density on the surface of the SMIWs with 5.5 mm in height were found to approach 2159 and 2239 mT, respectively. And the minimum value of the magnetic flux density around 355 um of SMIWs was found to approach 265 mT and this cross section is the surface of the SMIWs with 5.5 mm in height. The simulation result shows that the high magnetic field gradient can guide the magnetic microrobot from near the midpoint to the edge of the two SMIWs. At last, the magnetic microrobot could be guided in a methodical distribution process by dot-matrix magnetic flux density with a powerful magnetic flux density and high magnetic field gradient, which depended on the array layout of the soft magnetic wires to gather and discretize the magnetic field density.

The micromanufacture process of the magnetically guided device is as follow. Firstly, we pushed four square permanent magnets into a plastic fixing bracket which is made from the ultraviolet curable resin and fixed those magnets by the glue and the limit hole of the plastic fixing bracket. Subsequently, we pushed the central square permanent magnet into the hole, which is composed of the four permanent magnets, fixed those magnets by the glue. Four square permanent magnets were symmetrically combined in the plastic fixing bracket and their bottoms were flush with the bottom of the central square permanent magnet and the optical image is shown in Figure 5a. Secondly, the Sylgard 184-A (1 kg) and Sylgard 184-B (0.1 kg) were mixed in a 10:1 mass ratio to produce the Polydimethylsiloxane (PDMS). These reagents were used to fabricate the PDMS fixing module of the magnetically guided device and were purchased from Dow Corning Co, Ltd. The PDMS fixing module with a 9 × 9 homogeneous microporous array was fabricated via the injection molding method. The pore diameter, pore height and separation distance between the micropores were 200, 5500 and 500 μm, respectively. And then, the eighty-one SMIWs of the magnetically guided array were inserted into the micropore of the PDMS fixing module, which had a height of 5.5 mm and the optical image is shown in Figure 5b. Finally, five square permanent magnets with a plastic fixing bracket were placed under the petri dish and a rectangular area (4.4 mm × 4.4 mm) was selected to place the SMIWs above on the center of the central square permanent magnet. The optical image of the magnetically guided device is shown in Figure 5c. Therefore, the magnetic microrobot could be guided on the surface of a magnetically guided device and manipulated into desired pattern structures by the dot-matrix magnetic flux density.

### 3.2. Microfabrication of The Magnetic Microrobot

The magnetic nanoparticle within the alginate solution enabled the controllability of the magnetic microrobot by the rapid magnetic response. It was taken along with the magnetic microrobot and could be quickly captured with a magnetic field in the water. The magnetic microrobot was microfabricated by the cross-linking reaction inside the microfluidic device and the morphological structure of the magnetic microrobot was flexibly adjusted via precise control of the volumetric flow rate. The appropriate volumetric flow rate of the related solutions was accomplished by controlling three syringe pumps, which was the key to the morphology design of the magnetic microrobot.

Firstly, we formulated sodium alginate 80/120 powders (0.25 g) and dextran (200,000 molecular weight) powders (2 g) with ultrapure water to form 1.25% (w/v) alginate solution (20 mL) and approximately 10% (w/v) buffer solution (20 mL), respectively. We mixed calcium chloride powders (1.11 g) and dextran powders (2 g) with ultrapure water to form the 0.5 M gelation solution (20 mL). These agents were used to synthesize the microrobot by a cross-linking reaction inside the microfluidic device and were purchased from Wako Pure Chemical Industries, Ltd. (Japan). We formulated FeSO_4_ (1.39 g) and FeCl_3_ (2.7 g) powders to form the magnetic nanoparticles via the chemical co-precipitation method and formulated magnetic nanoparticles (0.05 g) into the alginate solution (10 mL) to form homogeneous magnetic alginate solution.

Secondly, we pulsed the buffer solution at 1000 μL/h into the microfluidic device and dredged the microchannels. Then, we set the volumetric flow rate of the buffer solution to 500 μL/h and this solution could be used to balance the viscosity of the solutions and to moderate the gelation speed. And we pulsed the buffer solution at 2000 μL/h into the microfluidic device to prevent blockage during the cross-linking reaction process, which prevented residues being left in the microchannels following the experiment.

Thirdly, we set the volumetric flow rate of the magnetic alginate solution to 500 μL/h to ensure that the magnetic microrobot had a sufficient width. After five minutes, the magnetic alginate solution and the buffer solution were stable. Then, we pulsed the gelation solution at 2500 μL/h into the microfluidic device to ensure the magnetic microrobot had an appropriate toughness, elasticity and width. The various solutions with related volumetric flow rates were set on the three syringe pumps is shown in Figure 6a and the various solutions were pulsed into the microfluidic device is shown in Figure 6b.

Finally, we smoothly spun the magnetic microrobots with appropriate morphological structure from the microfluidic device and the width of the microrobot was close to 200 μm. The microscopic image of the magnetic microrobots is shown in Figure 7a,b. (The verification experiment is shown in Appendix A—Rapid magnetic response of the magnetic microrobot.)

### 3.3. Contactless Micromanipulation of Magnetic Microrobot

We encapsulated the magnetic nanoparticle inside the magnetic microrobot in order to quickly capture the microrobot by the magnetic field. Therefore, the magnetically guided micromanipulation of the magnetic microrobot in liquid environment is as follows:

Firstly, we smoothly spun the magnetic microrobot from the microfluidic device and paved it on the side of the magnetically guided device which is fixed by an acrylic sheet with 3 mm thickness in the water and adjusted the spatial posture of the micro permanent magnet by movement control of the magnetically guided manipulator. When the micro permanent magnet went down the altitude which with 7.5 mm above the liquid level, the magnetic microrobot was not captured by the magnetically guided manipulator due to the long distance and it is shown in Figure 8a.

Secondly, we controlled the micro permanent magnet continuing descent and the head of the magnetic microrobot was captured by the magnetic field of the magnetically guided manipulator. The altitude is 5.5 mm above the liquid level and it is 10 mm above the magnetic microrobot. And then, we lifted up the head of the magnetic microrobot by upward movement of the magnetically guided manipulator and it is shown in Figure 8b.

Thirdly, when the head of the magnetic microrobot reached the liquid level, we continued to lift the magnetically guided manipulator up. Miraculously, the head of the magnetic microrobot broke away from the constraint of the magnetic force due to the liquid surface tension and gravity. And then, the head of the magnetic microrobot fell down due to gravity and it is shown in Figure 8c.

Therefore, this magnetically guided micromanipulation cooperate with the magnetically guided device can be used to accurately distribute the magnetic microrobot in the complex environment and the hardcore of the micromanipulation robot system is shown in Figure 8d.

### 3.4. Accurate Creation of Artistic Patterns

We applied the magnetically guided micromanipulation method to accurately create the artistic patterns with the magnetic microrobots in the water as follows:

Firstly, we smoothly spun the magnetic microrobot from the microfluidic device and paved it on the side of the magnetically guided device. We lifted up the head of the magnetic microrobot with the magnetically guided manipulator and the upward movement is terminated when it reached the reasonable height. Secondly, the magnetic microrobot was guided to the specific location above the magnetically guided device by the X-Y horizontal movement of the magnetically guided manipulator. We captured and guided it to the target location by employing the dot-matrix magnetic flux density. Thirdly, we lifted up the head of the magnetic microrobot with the magnetically guided manipulator and when it reached the liquid level, we continued to lift the magnetically guided manipulator up. the head of the magnetic microrobot broke away from the constraint of the magnetic force due to the liquid surface tension and gravity. And then, the head of the magnetic microrobot fell down due to gravity and reached the surface of the magnetically guided device. The most important and the most difficult of our method is to fix the head of the magnetic microrobot and other parts can be used in a similar way. The middle of the magnetic microrobot is controlled to up and down and it is shown in Figure 9a,b. Furthermore, we piecewise manipulated the middle of the magnetic microrobot by dragging the proper position of its front and the path planning is like fitting a circle with countless tangents.

Therefore, we spatially organized the magnetic microrobot into desired places and accurately created two helicoid patterns with one magnetic microrobot, respectively. And we achieved those artistic patterns through the precise control of the magnetically guided manipulator combined with the orderly distribution of the magnetically guided device. The clockwise helicoid pattern with one magnetic microrobot is shown in Figure 10a and the anticlockwise helicoid pattern with one magnetic microrobot is shown in Figure 10b.

Furthermore, we accurately created a circular pattern with one magnetic microrobot and this pattern is shown in Figure 11a. And we accurately created the rape flowers pattern with two petals based on the circular pattern created with one magnetic microrobot and this pattern with two magnetic microrobots is shown in Figure 11b.

Subsequently, we accurately created a bowknot pattern with one magnetic microrobot and this pattern is shown in Figure 11c. And we accurately created the Chinese knotting pattern with three magnetic microrobots based on the bowknot pattern created with one magnetic microrobot and this pattern is shown in Figure 11d. And then, we accurately created a four-leaf clover pattern with one magnetic microrobot and this pattern is shown in Figure 11e. We also accurately created the galsang flower pattern with eight petals based on the four-leaf clover pattern created with one magnetic microrobot and this pattern with two magnetic microrobots is shown in Figure 11f. Finally, we accurately created a “number 1” pattern with one magnetic microrobot and the pattern is still steady without willful drifting when we stir the liquid and inject bubbles with a micropipette. (The verification experiment is shown in Appendix A—Stability test of the number pattern.) During the experiment of the micromanipulation of the magnetic microrobot in water, the liquid keeps shaking but the pattern structure without devastating collapse. Therefore, the experimental results show accuracy and flexibility of the magnetically guided micromanipulation method by created the diverse artistic patterns in a liquid environment.

## 4. Conclusions

Herein, a method to accurately create diverse artistic patterns in a liquid environment was reported via organizing a magnetic microrobot onto a magnetically guided device by magnetically guided micromanipulation that depends on the precise control of the magnetically guided manipulator cooperated with the orderly distribution of the magnetically guided device. First of all, a magnetically guided device is developed to change the space distribution of magnetic field of the conventional permanent magnet and the powerful magnetic flux density and high magnetic field gradient that were generated depended on the symmetrical combination of 5 square permanent magnets and the array layout of 81 soft magnetic wires with 5.5 mm height to add stability to the pattern structure. Furthermore, the morphology of the magnetic microrobot was adjusted with precise control of the volumetric flow rate to give it flexibility and magnetic nanoparticles were used along with the microrobot to enable the rapid magnetic response that gave it controllability. Then, the spatial posture of the magnetic microrobot was precisely controlled by the magnetically guided manipulator without destructive manipulation and the microrobot was released from the magnetic force by the influence of surface tension and gravity. Subsequently, the artistic fashions with the magnetic microrobots were distributed via dot-matrix magnetic flux density by the magnetically guided device to avoid fluid disturbance. Finally, we created the artistic patterns in the liquid environment and established the accuracy and flexibility of the proposed method. This magnetically guided micromanipulation will provide a new way for personalized functional scaffold construction in tissue engineering and it has capable to provide a new method for targeted drug delivery or gastrointestinal disease examination in precision medicine application.

## Figures and Tables

**Figure 1 micromachines-11-00697-f001:**
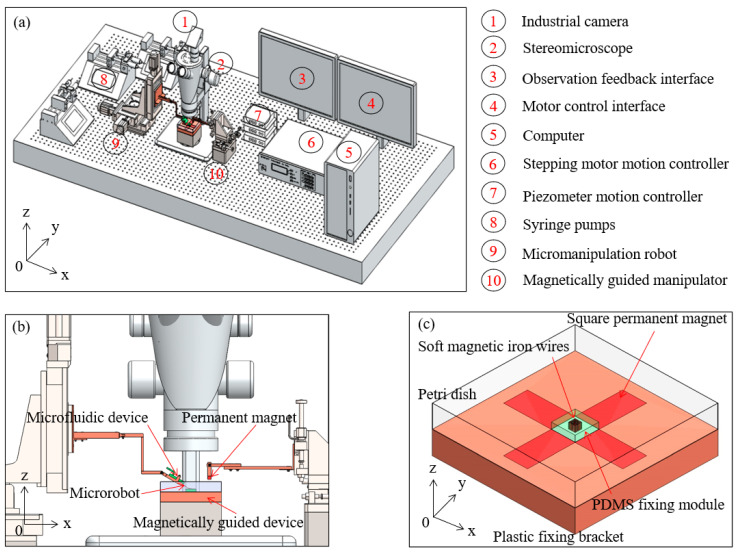
(**a**) The schematic diagram of the micromanipulation robot system; (**b**) The partial enlarged drawing of the mechanical subsystem; (**c**) The schematic diagram of the magnetically guided device.

**Figure 2 micromachines-11-00697-f002:**
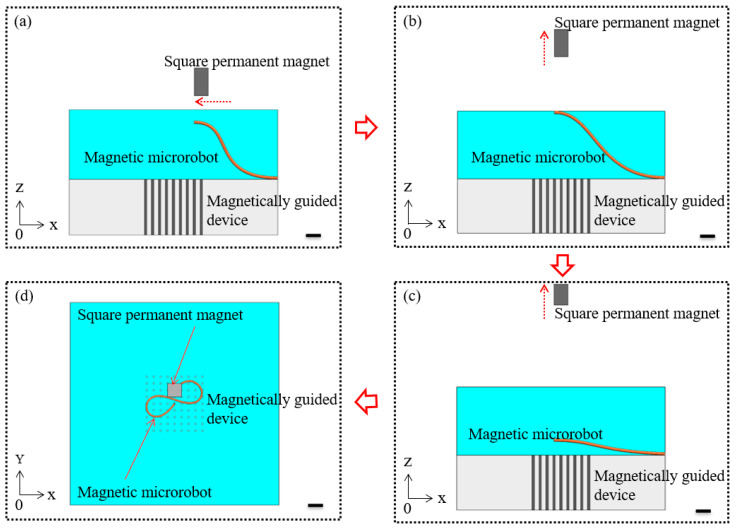
(**a**) The magnetic microrobot was guided to the specific location; (**b**) The magnetic microrobot was lifted up to the liquid level; (**c**) The magnetic microrobot was broken away and dropped; (**d**) The magnetic microrobot was captured and guided to the desired places to form an artistic pattern. Scale bars are 500 μm.

**Figure 3 micromachines-11-00697-f003:**
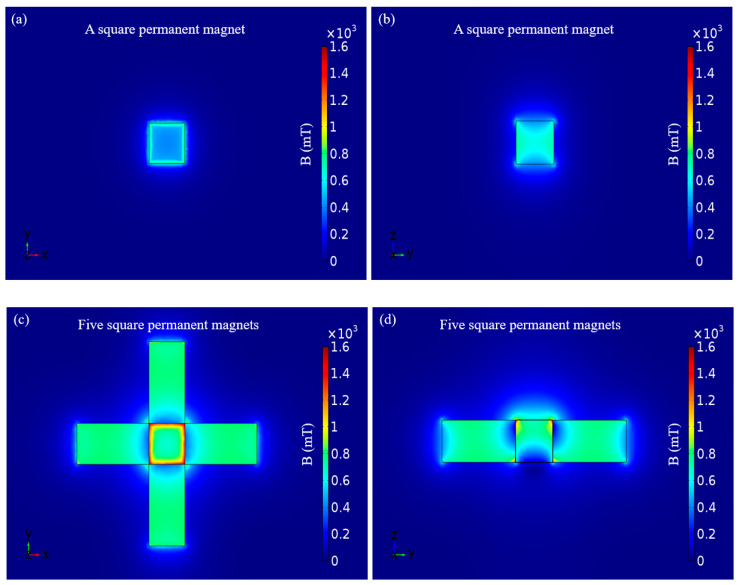
Simulation results of the square permanent magnet. (**a**) Vertical view of the magnetic flux density of the surface of the square permanent magnet; (**b**) Sectional view of the magnetic flux density which through the center of the square permanent magnet; (**c**) Vertical view of the magnetic flux density of the surface of the central square permanent magnet, which has four surrounding square permanent magnets; (**d**) Sectional view of the magnetic flux density which through the center of the central square permanent magnet and two centers of two symmetrically combined square permanent magnets.

**Figure 4 micromachines-11-00697-f004:**
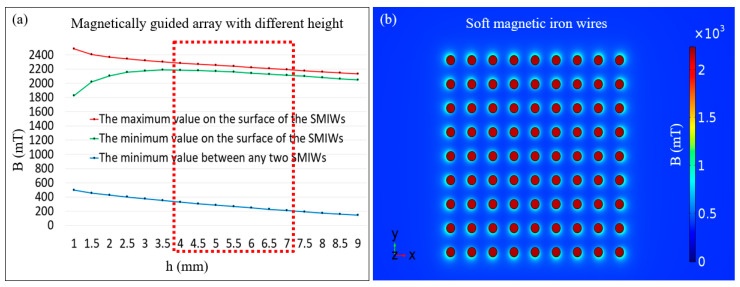
(**a**) Simulation results of the magnetic flux density above the surface of the soft magnetic iron wires (SMIWs) of the magnetically guided array with different height and the SMIWs with same height for every magnetically guided array; (**b**) Simulation result of the magnetic flux density above the surface of the magnetically guided device with the SMIWs 5.5 mm in height.

**Figure 5 micromachines-11-00697-f005:**
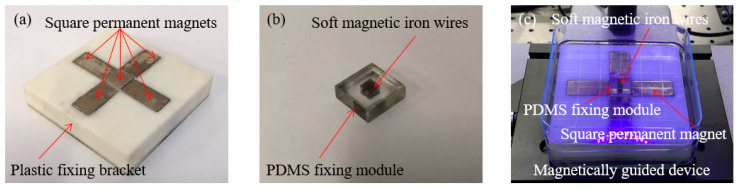
(**a**) Four square permanent magnets were symmetrically combined in the plastic fixing bracket and their bottoms were flush with the bottom of the central square permanent magnet; (**b**) Eighty-one SMIWs of the magnetically guided array with 5.5 mm height were inserted into the micropore of the polydimethylsiloxane (PDMS) fixing module; (**c**) Optical image of the magnetically guided device.

**Figure 6 micromachines-11-00697-f006:**
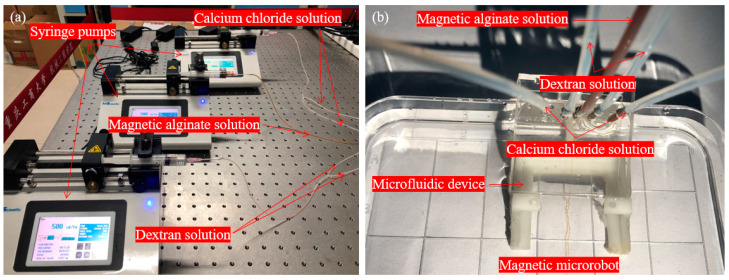
(**a**) The various solutions with related volumetric flow rates were set on the three syringe pumps; (**b**) The various solutions were pulsed into the microfluidic device.

**Figure 7 micromachines-11-00697-f007:**
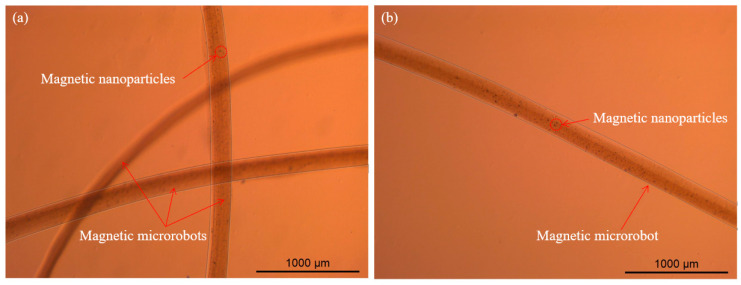
The microscopic image of the magnetic microrobots with appropriate morphological structure. (**a**) Three magnetic microrobots; (**b**) One magnetic microrobot which encapsulated the magnetic nanoparticles. The scale bar is 1000 um.

**Figure 8 micromachines-11-00697-f008:**
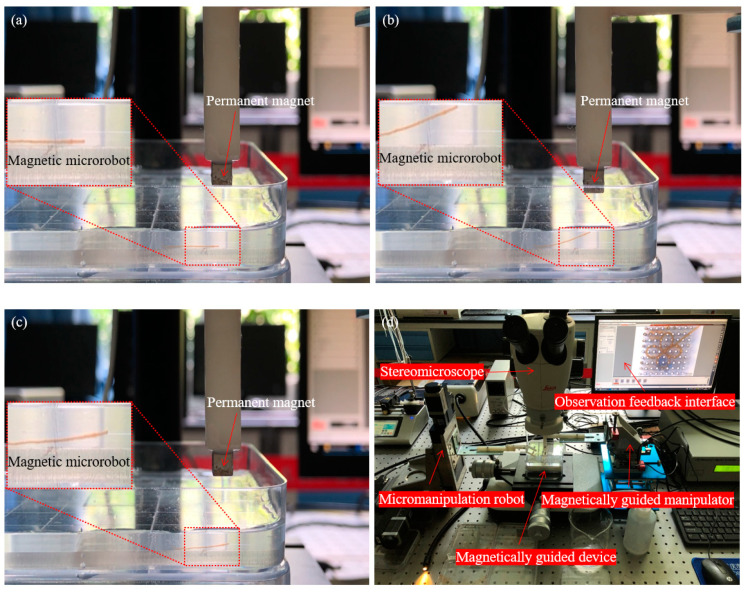
The magnetically guided micromanipulation of the magnetic microrobot in liquid environment; (**a**) The magnetic microrobot was not captured by the magnetically guided manipulator due to the long distance; (**b**) The head of the magnetic microrobot is lifted up by upward movement of the magnetically guided manipulator; (**c**) The head of the magnetic microrobot fell down due to gravity; (**d**) The hardcore of the micromanipulation robot system.

**Figure 9 micromachines-11-00697-f009:**
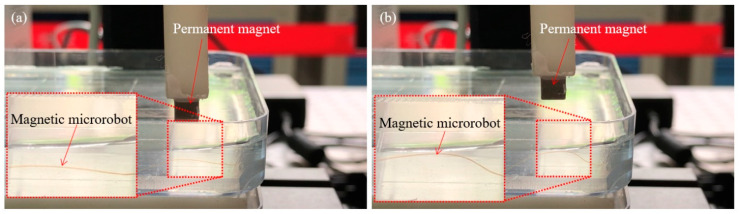
The middle of the magnetic microrobot is controlled to up (**a**) and down (**b**).

**Figure 10 micromachines-11-00697-f010:**
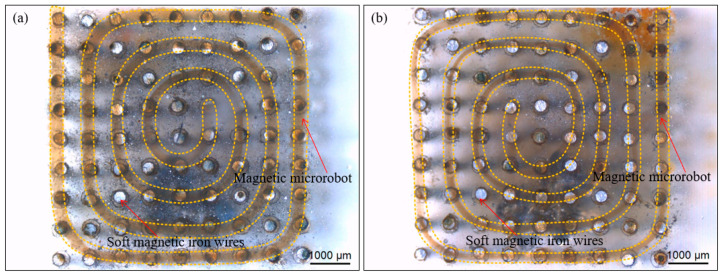
Images of the results of two helicoid patterns by magnetically guided micromanipulation of one magnetic microrobot. (**a**) The clockwise helicoid pattern with one magnetic microrobot; (**b**) The anticlockwise helicoid pattern with one magnetic microrobot. Scale bars are 1000 um.

**Figure 11 micromachines-11-00697-f011:**
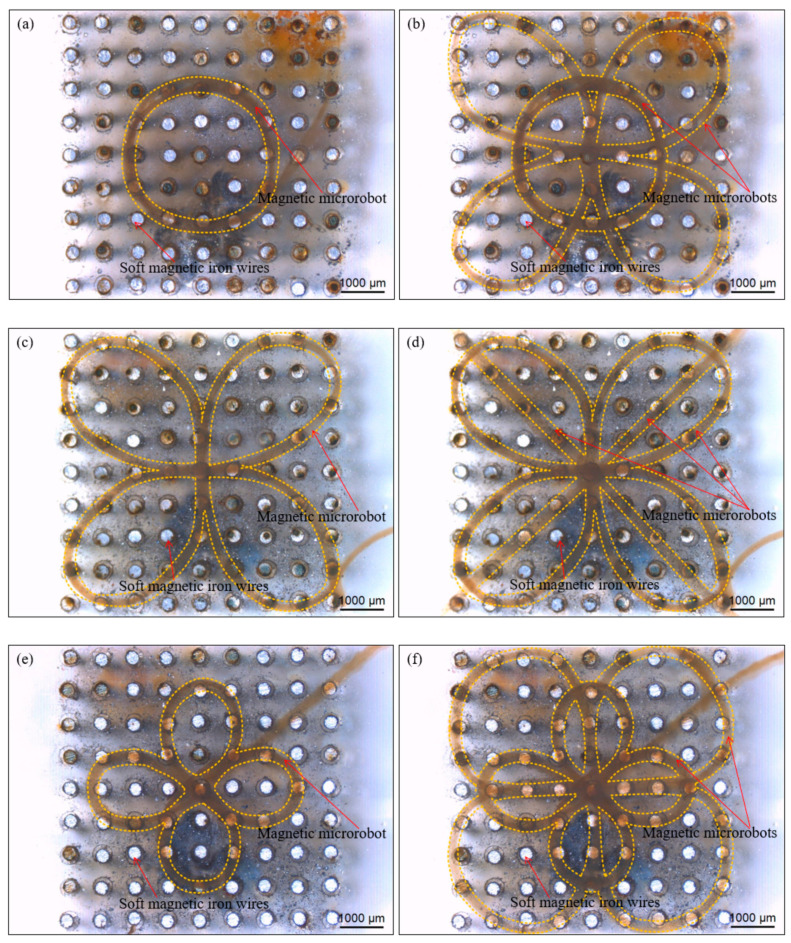
Images of the results of the accurate creation of the diverse artistic patterns by magnetically guided micromanipulation of the magnetic microrobot. (**a**) The circular pattern with one magnetic microrobot; (**b**) The rape flowers pattern with two magnetic microrobots based on the circular pattern; (**c**) The bowknot pattern with one magnetic microrobot; (**d**) The Chinese knotting pattern with three magnetic microrobots based on the bowknot pattern; (**e**) The four-leaf clover pattern with one magnetic microrobot; (**f**) The galsang flower pattern with two magnetic microrobots based on the four-leaf clover pattern. Scale bars are 1000 um.

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
