# Peer review of "Magnetically Guided Micromanipulation of Magnetic Microrobots for Accurate Creation of Artistic Patterns in Liquid Environment"

_micromachines, 2020, doi:10.3390/mi11070697_

Round 1
Reviewer 1 Report
The authors present a method to construct patterns in a fluidic environment using a magnetic fixturing design. The introduction is missing many citations from the magnetic manipulation community and could be enhanced. The work is new in its ability to manipulate several flexible magnetic ropes in a non-contact way to create shapes. The bulk of the study is focused on the hardware developed, but not the forces necessary, which makes the study reproducible but not extendable. Please see the following specific comments for areas of improvement.
- The authors are missing a large body of on magnetic manipulation in their literature review. I suggest starting with a recent review to help find additional relevant sources. https://doi.org/10.1146/annurev-control-081219-082713
- Their exist many grammatical and usage issues that prevent understanding. For example line 251: "we programmatic sited the volumetric flow rate" does not have whatever meaning you intended.
- It would be helpful to the reader to better explain what you mean by dot-matrix magnetic flux density in the introduction where it is first presented. Having finished the paper, I understand the concept, but it did not become clear to me until 2/3 the way through.
- How much force is necessary to pull the magnetic thread through the fluid? What is the longest thread you can maneuver?
- Can you manipulate threads from their middle instead of only the head to readjust?
- The resolution of your patterns appears to be the spacing of the magnetic studs, how fine of a stud-pattern can be made before their cross-talk prevents your dot-matrix fixturing of the thread?
- What is the tightest bend-radius you can achieve?
- Can you cross threads into knots or are they limited to purely planar patterns?
- How long does the arrangement take?
- What is the maximum speed you can move threads at?
- I notice corrosion beginning on your studs in Figure 10 (corner of a for example). How long does the magnetic pattern hold in solution given this corrosion?
- How stable are the shapes once patterned, that is how much disturbance force can they reject given the magnetic trap created by the magnetized wires?
Author Response
The authors would like to express our sincere appreciation for your beneficial comments and for the efforts on helping us to improve the quality and presentation of our manuscript. The attachment is the authors' responses to your comments.
Reviewer 2 Report
In this work, the authors proposed an interesting method to create artistic patterns in fluid, using magnetic wires. A new magnetic actuation setup is designed as well, which consists of a permanent magnetic guider, a magnetic based consists of five permanent magnets and a PDMS-based module embedded with soft magnetic iron wires. Overall, this work is interesting, and it would attract broad readership. Meanwhile, the authors should consider addressing the following issues before publication.
- Some critical concepts are confusing. For example: magnetic microrobots/magnetically guided robot. I recommend the authors to rename the items for better understanding.
- In the introduction, the motivation is not sufficiently clear. The authors are recommended to further discuss the importance and necessity of developing this technique.
- Figure 6 is not clear, especially Figure 6a. The wires are entangled with each other, which makes the figure confusing. A supportive schematic drawing can be considered to improve the quality. In Figure 6b, the shape of the “magnetic microrobot” is not satisfying. I understand the microrobot is actually a soft magnetic wire, so is it possible to present a better image showing that? Misunderstanding should be avoided.
- The microrobots in Figure 8 are also not clear. Better to improve the contrast, or use zoom-in insets to highlight them.
- The English expression of this manuscript is recommended to take an intensive revision, many places may make readers hard to follow.
Author Response

(The authors gave the same response as above.)

Round 2
Reviewer 1 Report
The authors have revised their manuscript in line with the comments provided. It is unfortunate that some of the more technical information around system capability cannot be assessed at this time, but the technique is unique enough that a this proof-of-principle demonstration has its own merit. Their remains usage and grammar issues that can be handled by a copy editor.